# Factors that motivate men who have sex with men in Berlin, Germany, to use or consider using HIV pre-exposure prophylaxis—A multi-methods analysis of data from a multicentre survey

**Matthew Gaskins** *, **Mary Katherine Sammons, Frank Kutscha, Alexander Nast, Ricardo Niklas Werner**

Division of Evidence-Based Medicine (dEBM), Department of Dermatology, Venereology and Allergy, Charité–Universitätsmedizin Berlin, Corporate Member of Freie Universität Berlin, Humboldt-Universität zu Berlin, and Berlin Institute of Health, Berlin, Germany

* matthew.gaskins@charite.de

## Abstract

### Background

While our knowledge of what motivates men who have sex with men (MSM) to use HIV pre-exposure prophylaxis (PrEP) has grown in recent years, quantitative survey-based studies have not asked MSM explicitly to name their motivations. We did so using a qualitative open-ended question and aimed to categorise the responses and explore whether these were related to where MSM were located along a conceptual continuum of PrEP care.

### Methods

In a multicentre survey examining knowledge and use of PrEP among MSM in Berlin, Germany, we additionally asked an open-ended question about motivations for using or considering PrEP. Data were collected from 10/2017-04/2018. One researcher developed a thematic framework deductively from the literature and another did so inductively from the free-text data, and a merged framework was used to code responses independently. We used Fisher's exact test to assess whether the frequency of motivations differed significantly between respondents using or considering PrEP.

### Results

Of 875 questionnaires, 473 were returned and 228 contained a free-text response. Motivations in the following categories were reported: (1) Safety/protection against HIV (80.2% of participants, including general safety; additional protection to condoms), (2) Mental well-being and quality of life (23.5%, including reduced anxiety; better quality of life), (3) Condom attitudes (18.9% intent not to use condoms), (4) Expectations about sexuality (14.4%, including worry-free sex or more pleasurable sex, with explicit mention of sex or sexuality),

**Data Availability Statement:** All relevant data are within the paper and its Supporting Information files.

**Funding:** The authors received no specific funding for this work.

**Competing interests:** The authors have declared that no competing interests exist.

(5) Norms/social perspectives (0.8%). The difference in frequencies of motivations between those using or considering PrEP was not statistically significant.

## Conclusions

Safety and protection against HIV, particularly having additional protection if condoms fail, were the most common motivations for using or considering PrEP, followed by mental well-being and quality of life. Many respondents reported several motivations, and responses overall were heterogeneous. This suggests that approaches to increase PrEP uptake that focus exclusively on its effectiveness in preventing HIV are unlikely to be as successful as a holistic approach that emphasises multiple motivations and how these fit into the broader sexual and psychological health of MSM.

## Introduction

Men who have sex with men (MSM) continue to be the population most affected by HIV in the United States (US) and Europe, comprising 69% of new HIV diagnoses in the US [1] and 52% of new diagnoses (where the route of transmission was known) in the European Union (EU)/European Economic Area in 2018 [2]. An important and still relatively new tool to prevent HIV is pre-exposure prophylaxis (PrEP) with emtricitabine/tenofovir disoproxil fumarate, which was approved for once-daily use in at-risk populations in the US in 2012 and in the EU in 2016. However, despite evidence of its high efficacy and safety [3–17], uptake among those at high risk of HIV acquisition has been slow. As of late 2019, approximately 224,000 people in the US were estimated to have received a prescription for PrEP, representing only a small proportion of the 1.1 million individuals estimated by researchers at the US Centers for Disease Control and Prevention (CDC) to have indications for it based on data from 2015 [18–20]. This is unfortunate given that the substantial declines in HIV incidence among MSM in London [21], San Francisco [22] and New South Wales, Australia [23], observed in the past several years are thought to be due, at least in part, to PrEP alongside other crucial prevention strategies such as treatment as prevention (TasP) and the continued promotion of condom use. In Berlin, the decline in HIV incidence among MSM has been smaller [24], and affordable PrEP has been accessible through formal channels in Germany only since autumn 2017.

In Europe, where health policy makers and payers in many countries are still debating whether and how best to integrate PrEP into their publicly funded health systems and broader HIV prevention strategies [25–28], data on PrEP uptake are sparse. The most recent and comprehensive figures are from a 2019 study using data from the European MSM Internet Survey (EMIS-2017) to estimate the gap between self-reported PrEP use and expressed need for PrEP in the EU. Using this definition, the authors found that an estimated 17.4% of MSM, or 500,000 individuals, in the EU who were very likely to use PrEP were not able to access it [29]. At 12.6%, the PrEP gap for Germany in 2019 was smaller than the EU average [29], but not inconsiderable, and was similar to a finding of the first facility-based paper survey of PrEP use among MSM (N = 470) in Berlin in 2017/18, of which the current study is a part: in the survey, 12.9% of participants who reported never having used PrEP (who comprised 82.3% of the sample) strongly agreed with the statement that they would like to do so [30]. The most frequently reported barriers to PrEP use by these participants were worries about adverse effects, cost, not

having a doctor who prescribes it, and, despite a high awareness of PrEP, a lack of information on its pros, cons and proper use [30].

To improve the implementation of PrEP, a range of authors over the past decade have proposed and refined several models of a continuum or "cascade" of PrEP care [31–37]. Most of the earlier models are largely based on a biomedical approach that assumes HIV prevention is the main priority in a person's sexual decision-making and therefore evaluate this decision-making chiefly in terms of risk [38] and in a linear fashion. For example, the model proposed by Nunn et al. (2017) comprises nine steps ranging from identifying individuals at the highest risk of HIV to an end goal of retaining individuals in PrEP care [33]. In contrast, one of the more recent iterations of the PrEP care continuum, developed by Newman et al. (2018), follows a more holistic notion of personhood rooted in the results of their own qualitative analysis and the extensive anthropological and ethnographic literature on HIV prevention [37]. Their model considers the psychological, interpersonal, social and community phenomena that affect PrEP use and non-use, resulting in an augmented PrEP cascade that incorporates alternative decision-making, pathways (such as those with the end goal of intermittent use or discontinuation), and psychosocial impacts [37]. Regardless of the epistemological roots and approaches of these different models, however, what all of them share is that each stage in the proposed cascade represents a potential point of intervention to improve PrEP implementation among those for whom PrEP is a desired and viable option [32, 33]. At the same time, each of these steps is characterised by unique but overlapping sets of barriers and enablers that must be considered if we are to understand more fully how and why individuals move from one step to another–an understanding that is crucial to developing interventions to improve the implementation of PrEP, and HIV prevention strategies more generally, that are efficient, effective and, ideally, support the broader sexual and psychological health of MSM.

Since 2012, evidence on the structural, sociocultural and behavioural barriers to PrEP implementation has grown considerably. These are manifold and include cost and lack of insurance coverage (e.g., [29, 39–43]), access to providers willing to prescribe PrEP (e.g., [18, 41, 44]), concerns among providers about increased sexual risk compensation (e.g., [29, 39]), but also, among those using or considering PrEP, concerns about side effects (e.g., [18, 39, 42, 43, 45]), stigma and discrimination (e.g., [18, 39, 43, 46–48]), and low perceived risk of infection (e.g., [18, 49–53]). Until recently, one significant gap in the literature was the lack of data on individuals' motivations for taking PrEP. Indeed, in their comprehensive early review of research on the acceptability of PrEP and treatment as prevention (TasP), Young and McDaid (2014) identified motivations for PrEP use and adherence as one of the main areas in which further research was needed in order to identify "when, where and for whom PrEP and TasP would be most appropriate and effective" [45, p. 195].

A number of researchers, particularly in anthropological, ethnographic and other primarily qualitative lines of enquiry, have since stepped up to the plate to examine this topic. By exploring the subjective experiences of MSM in their navigation of PrEP, these authors have identified, mostly through in-depth interviews or in focus groups, numerous and varied motivations for using (or not using) PrEP that call into question the usefulness of biomedical approaches to PrEP implementation that de-emphasise psychosocial phenomena. In many cases, the motivations identified in these studies have stemmed from powerful affective experiences, such as being freed from sometimes decades-long, often cyclical anxiety about HIV infection [37, 38, 54, 55]; feeling empowered, able to make one's own informed choices, in control or autonomous [37, 56]; and feeling less fear and shame in relation to pre-existing high-risk sexual behaviours alongside greater sexual satisfaction and intimacy [57, 58]. The qualitative literature also reports motivations related to fearing or experiencing PrEP-related stigma, such as beliefs that PrEP is only for those who are highly promiscuous [56, 59]; being labelled a

"Truvada whore" (or similar) in lesbian, gay, bisexual, transgender, intersex and other gender diverse (LGBTI+) media, or by peers or even friends [60]; encountering provider-level stigma, including judgemental behaviour about the decision to use PrEP [35]; or experiencing, as a PrEP user or non-user, increased pressure to engage in condomless anal intercourse [37, 58, 61].

Among the studies that have explored motivations using survey-based quantitative methods, many have approached the subject in an indirect fashion using, for instance, correlation or regression analysis to identify factors associated with PrEP use, non-use, willingness to use PrEP or PrEP acceptability. A recent example is an analysis of data from a national, online, open-prospective observational study among MSM in Australia by Keen et al. (2020), who found that PrEP use was independently associated with lower levels of HIV anxiety among PrEP-eligible men [54]. Another example is the correlation between lower perceived HIV risk and PrEP non-use or PrEP discontinuation, which has been reported in large number of studies to date (e.g., [31, 51–53, 56, 58]). Furthermore, in a cross-sectional study of 164 HIV-negative MSM in HIV seroconcordant negative primary partnerships who were not taking PrEP, Gamarel et al. (2015) found that age, education and intimacy motivations for condomless sex were associated with PrEP adoption intensions in a multivariate model based on data elicited using interview-administered survey questions and assessment scales. Clearly, all of these factors can be interpreted as motivations for considering or using PrEP, but in none of the examples mentioned above did the survey participants actually describe them as such. Indeed, among this latter stream of the quantitative literature, we were unable to identify any surveys that explicitly asked MSM to name or describe their motivations for taking PrEP, whether by providing them with a list of pre-specified motivations or the possibility of writing a free-text response.

While in-depth qualitative interviews and focus groups are probably the ideal way to explore the complex affective phenomena in this area of enquiry, there are several possible methodological advantages to eliciting such qualitative information using a self-administered survey tool: positioning an open-ended question amidst many closed questions might invite quick, spontaneous responses, and being open-ended it would, moreover, not be influenced by pre-specified categories or subjects (as would be the case with multiple choice items and/or the use of Likert scales) [62, 63]. Also, compared to data from a qualitative interview, the responses to such a question are more amenable to being coded and then analysed using a frequentist approach, and correlations with responses to the closed questions in the survey can be explored. The results can subsequently be reflected against those of previous, more in-depth qualitative research, with the aim of confirming, refuting, complicating or contextualising these. Moreover, whereas some previous work on different models of the PrEP care continuum has located participants of various cohorts along the respective cascade, to our knowledge no studies to date have attempted to identify whether the motivations of participants at one location in a PrEP care continuum might differ significantly from those at another location in the continuum. Lastly, it should also be noted that the focus of much of the qualitative and quantitative research on PrEP motivations to date has been on the US context, as well as on Australia, Canada and the United Kingdom. Perhaps not surprisingly given the four-year lag in the approval of PrEP in the EU/EEA [64], data from this region are minimal in comparison, and Germany is no exception in this regard.

To help fill this gap in the literature, we conducted a multicentre, paper-based survey in late 2017/early 2018 of almost 500 MSM in Berlin with a self-reported negative or unknown HIV serostatus. In it, we explored participants' knowledge and use of HIV pre-exposure prophylaxis using mostly closed questions. The results of this part of the survey are reported elsewhere [30]. The survey also had an open-ended question in which we explicitly asked participants

who were using or considering PrEP to write what their main motivation was for doing so. In the current paper, we present the results of a multi-methods analysis of participants' free-text responses to this question. Our primary aim was to identify, describe and categorise the motivations of MSM in Berlin to use or consider using PrEP. Additionally, drawing upon results from the quantitative part of the survey, we sought to explore whether motivations differed between participants who had already started taking PrEP and those who were still considering it. Such information could help clinicians, sexual health counsellors and policy makers develop and distribute targeted information and advice to individuals at several steps of the PrEP care continuum and thereby improve the implementation of PrEP in Germany and beyond.

## Materials and methods

### Study design, ethics approval and informed consent

We used an anonymous, self-administered, paper-based questionnaire, available in German and English, to conduct a multicentre, cross-sectional survey of MSM in Berlin. The local institutional ethics committee at Charité–Universitätsmedizin Berlin approved the study protocol, the format and content of the questionnaire, and the information cover sheet for participants (EA1/162/17; 28/09/2017). All participants gave informed verbal consent to their physician or sexual health counsellor in English or in German before filling in the questionnaire. Participation was voluntary, and neither the centres nor the participants received any incentives to take part. The survey period was from 1 October 2017 to 2 April 2018. The questionnaires and information cover sheet can be accessed in full, in both English and German, in the publication of the quantitative study results by Werner et al. (2018) [30].

### Sampling methods, participants and settings

To be eligible, participants had to identify as male, be aged 18 years or older, report having sex with men, and have a self-reported negative or unknown HIV status. We collected data from a range of settings, namely all four non-municipal counselling and testing centres for the sexual health of LGBTI+ people, as well as six large HIV specialist practices in Berlin. The former are non-commercial walk-in centres funded by the state government of Berlin and through private donations. They offer a range of low-barrier services, including community outreach and anonymous counselling on legal, immigration and health issues for LGBTI+ adults and adolescents, as well as affordable testing for STIs, including HIV. They are not permitted by law to prescribe medication. All clients attending the walk-in centres for STI screening or counselling were offered participation in the study by the centre staff.

HIV specialist practices in Germany offer a full range of generalist and sexual health services to LGBTI+ people regardless of whether they are living with HIV, and indeed can serve as the designated GP/family doctors for these individuals. The practices are owned and staffed by physicians, and consultations usually require an appointment, but walk-ins are possible. In total, 11 such practices from seven districts in Berlin were invited to participate in the study. We chose these purposively based on their geographic spread across Berlin and our awareness that they had been willing to take part in previous research projects related to HIV and other STIs. In these practices, we asked doctors to include every eligible patient, irrespective of the reason for the consultation, in a consecutive manner.

During the data collection period (1 October 2017 to 2 April 2018), PrEP was available in Germany only by means of a private prescription, which could be written by any licensed doctor. About half way through this period, the price of a month's supply of generic PrEP in Germany fell from approximately 600 euros to as low as 50 euros [30]. The relevant guideline for identifying at-risk patients and prescribing PrEP is the German and Austrian Guideline on

HIV pre-exposure prophylaxis [65], but its recommendations are not binding. The German system of statutory health insurance began to cover the costs associated with PrEP in September 2019.

All of the HIV and STI testing and counselling centres in Berlin (n = 4) participated in the study, as did six of the 11 HIV specialist practices we had invited. Of the 875 questionnaires distributed by the participating centres, 473 were returned (response rate 54.1%). Three participants were excluded because they reported in the questionnaire that they were living with HIV. Of the remaining 470 questionnaires, 259 contained a free-text response to the question about participants' main motivation for using PrEP or considering its use. Thirty one of these responses were considered invalid because the participants had disagreed/strongly disagreed with the statement "I would like to take PrEP" (n = 30) or because they responded "no" to the question whether they had ever used PrEP but had indicated in their response to the statement "I would like to take PrEP" that it was not applicable to them because they were already taking it (n = 1). This resulted in a sample of 228 questionnaires with a valid free-text response for analysis.

## Questionnaire format and content

The questionnaire was two pages in length and consisted mostly of closed, multiple-choice questions on participants' knowledge and use of HIV pre-exposure prophylaxis, as well as demographic data and information on their sexual behaviour and HIV risk. Additionally, the survey had the following open-ended question: "If you are considering or already using PrEP, what is your main motivation for this?" The question gave space for responses of one to two sentences. A cover page gave a brief description of PrEP and explained the purpose of the survey.

## Qualitative data analysis

We defined our units of analysis as the individual motivations for using or considering PrEP rather than the free-text response as a whole. We applied a framework analysis approach to code each motivation, using high-level themes to categorise the data [66]. To help ensure that the analysis was not driven primarily by our theoretical interests and analytic preconceptions, we took a combined deductive and inductive approach to developing the coding framework [66, 67]: One researcher (RNW) created a set of categories and subcategories deductively based on the findings of a structured search of the literature for existing frameworks and models for characterising motivations for PrEP use, whereas a second researcher (MS) created a separate set of categories and subcategories inductively based on a pilot coding and analysis of a randomly selected subset of 75 free-text responses. These two classification systems were subsequently merged by the two researchers through discussion, with discrepancies and disagreements resolved by consensus in a series of meetings facilitated by a third researcher (MG).

The first researcher (RNW), who created a set of categories and subcategories deductively, took the following approach: To identify studies or reviews reporting a classification system or listing categories of motivations to use PrEP, he searched MEDLINE on 4 April 2018 using the search terms "motivation" and "PrEP", with time limits set to the five years before this date. Categories identified in the literature were noted without further assessment of the quality of the studies. The search yielded 61 records whose abstracts and titles were screened for eligibility, yielding six records to be evaluated in full text [51, 58, 68–71]. Two of these studies were used to derive an initial categorisation system for motivations: Dubov et al. (2018) reviewed factors associated with PrEP uptake and categorised these factors along the Information-Motivation-Behavioural Skills model [51]. Their construct of motivation comprised four domains,

each of which is attributed various further subdomains. Frankis et al. (2016) analysed qualitative data from in-depth interviews to deduce motivations for potential PrEP use, including situations of increased need for protection from acquiring HIV [68].

The second researcher (MS), who developed a set of categories and subcategories inductively, took the following approach: first, the third researcher (MG) familiarised himself with the entire qualitative dataset by reading and re-reading all free-text responses to make a pragmatic assessment of the number of these that would need to be coded before saturation was reached (i.e., roughly one third of all valid responses). Subsequently, a list of randomly generated numbers was appended to the qualitative dataset, and the responses numbered between one and 75 were sent to the second researcher for pilot coding. Coding was undertaken in a manner that was both open (i.e., with codes being assigned to describe as many perspectives as possible, such as particular behaviours, intentions, values, beliefs and emotions) [66] and circular (i.e. looping or cycling repeatedly between the qualitative codes assigned to the data and the data itself, and recoding, considering emergent categories, and recategorising as needed) [72].

After the two classification systems had been merged, the final categories comprised: safety/protection against HIV, expectations about sexuality, mental well-being and quality of life, condom attitudes, and norms/social perspectives. Table 1 gives an overview of the categories

**Table 1. Overview of final coding framework and categorisation system with definitions (categories and subcategories listed in alphabetical order).**

| Category | Subcategory |
|---|---|
| **CONDOM ATTITUDES [51]** | **Desire or intent to engage in condomless sex** |
| This category comprises two subcategories for coding responses that refer to respondents' attitudes towards condom use. | Responses are coded in this subcategory if they refer to the desire or intent to engage in condomless sex. Negative attitudes towards condom use and reporting episodes of condomless intercourse have been described as being associated with the intention to use PrEP [73]. |
| | **Difficulties with condom use** |
| | Responses are coded in this subcategory if they mention difficulties with condom use as a motivation to use PrEP [55]. |
| **EXPECTATIONS ABOUT SEXUALITY** | **Expectations of more pleasureful sex or increased intimacy and closeness (when not using a condom)** |
| This category comprises two subcategories for coding responses that refer to respondents' expectations about sexuality while using PrEP. | Responses are coded in this subcategory if they include the expectation of more pleasureful sex, intimacy or closeness as a motivation to use PrEP, irrespective of whether the response refers to using condoms. Believing that condoms reduce intimacy and closeness and/or sexual pleasure is a factor that has been described as associated with the intention to use PrEP [37, 55, 58]. |
| | **Expectations of worry-free sex** |
| | Responses are coded in this subcategory if they refer to the expectation of worry-free or less worrisome sex as the motivation for using PrEP [37]. In contrast to the subcategory "Reducing anxiety, fear, or worries of being infected with HIV" under the category "Mental well-being and quality of life", responses here had to mention sex or sexuality explicitly. |
| **MENTAL WELL-BEING AND QUALITY OF LIFE** | **Desire for a healthy life** |
| | Responses are coded in this subcategory if they refer to the general desire to increase health or longevity, or to lead a healthy life. |
| This category comprises four subcategories for coding responses that refer to respondents' mental well-being or aspects of general health. | **Desire to increase quality of life or sexual/personal freedom** |
| | Responses are coded in this subcategory if they refer to the desire to increase quality of life, mental well-being, general health, or sexual or personal freedom as the motivation to use PrEP [37]. |
| | **Reducing anxiety, fear, or worries of being infected with HIV** |
| | Responses are coded in this subcategory if they include the desire to reduce anxiety, fear or worries about being infected with HIV [37, 38, 54, 55]. Unlike the subcategory "expectations of worry-free sex", this subcategory does not include responses that refer explicitly to the act of sex. |
| | **Reducing periods of anticipated regret** |
| | Responses are coded in this subcategory if they included the desire to decrease periods of anticipated regret or worries. The cognitive-based emotion of anticipated regret from engaging in HIV-risk behaviour has been described as an important determinant of the intention to use PrEP [51]. |
| **NORMS / SOCIAL PERSPECTIVES** | **Perceiving condomless sex / PrEP intake as a social norm** |
| This category comprises two subcategories for coding responses that refer either to perceptions of PrEP use as a social norm or that reflect upon PrEP use in terms of social or public health perspectives. | Responses are coded in this subcategory if the answer refers to perceptions of PrEP use as a social norm or the need to use PrEP as the only means of personal protection in a social environment that insists on condomless sex [37, 58, 61]. |
| | **Prevention altruism [51]** |
| | Responses are coded in this subcategory if the respondent refers to a general public health perspective of reducing the burden of HIV epidemics. General public health concerns have been described as a facilitator of engaging in safer sex practices [74]. |

*(Continued)*

**Table 1.** (Continued)

| Category | Subcategory |
|---|---|
| **SAFETY/PROTECTION AGAINST HIV** | **Additional protection against HIV** |
| This category comprises eight subcategories for coding responses that refer to protection against HIV or general safety considerations, as well as more specific aspects of protection or safety for oneself or for others. | Responses are coded in this subcategory if they reflect the respondent's wish to have additional protection against HIV, or additional safety or security, by using PrEP as a "backup preventive strategy". This has been described in the literature as a specific motivation to use PrEP [68]. Safer sex intentions have been shown to be linked with the motivation to use PrEP [73]. |
| | **Autonomy and self-empowerment in the protection against HIV** |
| | Responses are coded in this subcategory if they reflect the respondent's wish to protect himself from being infected with HIV using a method of protection that lies within his own responsibility and is not dependent on his partners' reliability or will to use condoms [37, 56]. |
| | **Being at self-perceived risk of HIV** |
| | Responses are coded in this subcategory if they reflect the respondent's general perception of being at risk of acquiring HIV due to specific circumstances, such as having sex with many casual partners or being in a relationship with a person living with HIV. Self-perceived risk of acquiring HIV has been described as a factor that may motivate individuals to seek preventive services. Considering oneself as being at risk of HIV infection has been shown to be correlated with self-perceived eligibility for PrEP use [31, 51–53, 56, 58]. |
| | **PrEP as an affordable way to protect against HIV** |
| | Responses are coded in this subcategory if they mention the affordability or cost of PrEP as an option to protect oneself from being infected with HIV. |
| | **Protecting partner(s) or relationship(s) from HIV infection** |
| | Responses are coded in this subcategory if they reflect the respondent's wish to protect his (sex) partners' health or well-being or if the answers included relationship-associated aspects. Attitudes towards using PrEP have been shown to be linked with considerations of protecting primary and/or outside partners [55, 73]. Concerns for the sexual partners' risk of acquiring HIV and general public health concerns have also been described as a facilitator of engaging in safer sex practices [74]. |
| | **Protection against HIV during periods of anticipated increased risk (e.g., recreational drug use, holidays)** |
| | Responses are coded in this subcategory if they reflect the respondent's wish to protect himself from being infected with HIV during specifically defined events (e.g., recreational drug use) or periods (e.g., holidays) that are accompanied by an anticipated increased risk of being infected with HIV. PrEP has been described as an option for situations in which regular patterns of sexual practice might be disrupted, such as holidays or in the event of alcohol and/or drug use [53, 68]. |
| | **Protection against HIV, prevention of HIV and general safety** |
| | Responses are coded in this subcategory if they reflect the respondent's general wish to protect himself from being infected with HIV, or his generally expressed need for safety, without mentioning specific circumstances or specifying aims beyond (his individual) protection. |
| | **Protection against HIV when not using condoms** |
| | Responses are coded in this subcategory if they reflect the respondent's wish to protect himself from being infected with HIV explicitly without having to use condoms. |

and subcategories in our coding system, providing definitions for each, as well as references to the pertinent literature.

Two researchers (RNW, MS) subsequently used this framework independently to identify, code and classify individual motivations in each free-text response (including the 75 responses used in the pilot coding). Meetings among all three researchers were held at regular intervals

during the coding and classification process to discuss potential emerging new categories or subcategories during the coding. Ultimately, no substantial changes to the framework were needed during this process, although we did clarify the definition for the subcategory "Expectations of worry free-sex" to require, in contrast to the subcategory "Reducing anxiety, fear, or worries of being infected with HIV", the explicit mention of sex or sexuality. Moreover, a subcategory "Desire to increase quality of life or sexual freedom" was broadened to include "sexual/personal freedom". Furthermore, even though the number of responses in the category "Norms / social perspectives" was minimal, we decided to retain it because of its prominence as a finding in the qualitative literature (e.g., [37, 58, 61]) and our desire not to take an overly frequentist approach to reporting the qualitative data. Lastly, while preparing the manuscript we decided to use the term "quality of life" instead of "general health" to describe the category "Mental health and quality of life" because we felt the term better captured the nature of the responses we had assigned to various subgroups in this category.

There was a high degree of agreement between the coding results of the two researchers (Cohen's Kappa = 0.837, 95% CI: 0.794–0.880). The third researcher (MG) reviewed all disagreements and resolved these with the other researchers through discussion.

## Quantitative data analysis

No formal sample size calculations were performed for the survey. Based on feasibility considerations, we aimed at collecting data from approximately 500 participants. Data were analysed using descriptive statistics and Fisher's exact test of independence to assess whether the frequency of motivations for PrEP, by category, differed in a statistically significant manner (alpha level 0.05) between the subset of the sample that reported using PrEP or having a history of its use and the subset that reported considering PrEP.

In our data set, we classified respondents who indicated no previous PrEP use, reported a motivation, and did not disagree or strongly disagree with the statement that they would like to use PrEP as considering PrEP use. Within all of the models of the PrEP care continuum developed to date, this group could be located at any of the steps before the step corresponding to PrEP initiation (e.g., "Initiate PrEP" in the model by Newman et al. (2018) [37], Stage 4b of "Stage 4: PrEP action and initiation" in the model by Parsons et al. (2018) [34], or "Step 7: Initiating PrEP" in the model by Nunn et al. (2017) [33]). Respondents who reported that they were currently using PrEP or had used it in the past (i.e. "Yes, but not on a regular basis", "Yes, I regularly use it before and after risk sex (as needed)", "Yes, I use it continuously"), reported a motivation, and did not disagree or strongly disagree with the statement that they would like to use PrEP were classified as using PrEP or having a history of its use. This group could be located, roughly, in "Stage 5: PrEP maintenance and adherence" of the model by Parsons et al. (2018) or at steps 8 ("Adhere to PrEP") or 9 ("Retention in PrEP Care") of the model by Nunn et al. (2017). However, because we included people with different PrEP trajectories (i.e., those who had used PrEP in the past but were not necessarily taking it at present), they would be more appropriately placed between the steps "Initiate PrEP" and "Retention", with the possibility of being located at the stages "Seasonal or intermittent use" or "Discontinuation" on the augmented PrEP cascade of Newman et al (2018) [37].

For any post-hoc pairwise comparisons, we applied a Bonferroni-corrected alpha level of 0.05 divided by the number of compared pairs to account for multiple testing. To avoid a purely frequentist approach and preserve the richness of our qualitative data when reporting examples of motivational factors given in response to the open question, we chose to cite whenever possible five motivations that summarised the overall body of motivations in each subcategory, whether responses in this subcategory were frequent or rare. In cases where fewer

or more than five motivations were sufficient or needed for this purpose, a different number of responses is cited. Statistical analyses were performed using Stata SE version 14.2.

## Results

### Demographic data, sexual behaviour and HIV risk of analysis sample

Among the 228 respondents who gave a valid free-text response to our question about motivation, 65 were using PrEP or had used it at some point in the past, and 163 were considering it. Their mean age was 36.4 years (SD 10.8, range 20–79). Table 2 shows these and other

**Table 2.** Demographic data and sexual risk behaviour of participants who answered the question about their motivation for using or considering PrEP.

| | Analysis sample | Participants using PrEP (or with history of its use) | Participants considering PrEP use |
|---|---|---|---|
| | N = 228 | N = 65 | N = 163 |
| **Age** | | | |
| Mean (SD) in years | 36.4 (10.8) | 33.9 (6.9) | 37.3 (11.8) |
| Min-max in years | 20–79 | 24–53 | 20–79 |
| Not stated | 9 | 5 | 4 |
| **Highest degree or level of school (N, %)** | | | |
| Primary education | 0 (0.0%) | 0 (0.0%) | 0 (0.0%) |
| Secondary education up to year 10* | 21 (9.2%) | 3 (4.6%) | 18 (11.0%) |
| Secondary educ. with apprenticeship | 11 (4.8%) | 1 (1.5%) | 10 (6.1%) |
| Secondary education up to year 12** | 41 (18.0%) | 7 (10.8%) | 34 (20.9%) |
| University degree | 151 (66.2%) | 50 (76.9%) | 101 (62.0%) |
| Not stated | 4 (1.8%) | 4 (6.2%) | 0(0.0%) |
| **Financial situation (N, %)** | | | |
| Not always have enough money | 19 (8.3%) | 7 (10.8%) | 12 (7.4%) |
| Enough money | 104 (45.6%) | 25 (38.5%) | 79 (48.5%) |
| More than enough money | 102 (44.7%) | 30 (46.2%) | 72 (44.2%) |
| Not stated | 3 (1.3%) | 3 (4.6%) | 0 (0.0%) |
| **Place of residence (N, %)** | | | |
| Berlin | 213 (93.4%) | 60 (92.3%) | 153 (93.9%) |
| Other city in Germany | 7 (3.1%) | 1 (1.5%) | 6 (3.7%) |
| Small town / rural area in Germany | 0 (0.0%) | 0 (0.0%) | 0 (0.0%) |
| Other country | 5 (2.2%) | 1 (1.5%) | 4 (2.5%) |
| Not stated | 3 (1.3%) | 3 (4.6%) | 0 (0.0%) |
| **Family origins (N, %)** | | | |
| Participant & both parents born in Germany | 132 (57.9%) | 29 (44.6%) | 103 (63.2%) |
| One parent born outside Germany | 17 (7.5%) | 7 (10.8%) | 10 (6.1%) |
| Both parents born outside Germany | 19 (8.3%) | 7 (10.8%) | 12 (7.4%) |
| Participant born outside Germany | 56 (24.6%) | 19 (29.2%) | 37 (22.7%) |
| Not stated | 4 (1.8%) | 3 (4.6%) | 1 (0.6%) |
| **Current HIV status (N, %)** | | | |
| HIV negative | 198 (86.8%) | 61 (93.8%) | 137 (84.0%) |
| Not sure | 24 (10.5%) | 0 (0.0%) | 24 (14.7%) |
| Not stated | 6 (2.6%) | 4 (6.2%) | 2 (1.2%) |
| **STI diagnosis in the past six months (N, %)** | | | |
| No | 168 (73.7%) | 37 (56.9%) | 131 (80.4%) |
| Yes | 57 (25.0%) | 25 (38.5%) | 32 (19.6%) |

*(Continued)*

**Table 2.** (Continued)

| | | Analysis sample | Participants using PrEP (or with history of its use) | Participants considering PrEP use |
|---|---|---|---|---|
| | | **N = 228** | **N = 65** | **N = 163** |
| | Not stated | 3 (1.3%) | 3 (4.6%) | 0 (0.0%) |
| **Role when having anal sex (N, %)** | | | | |
| | No anal sex | 7 (3.1%) | 0 (0.0%) | 7 (4.3%) |
| | Bottom only | 21 (9.2%) | 6 (9.2%) | 15 (9.2%) |
| | More bottom than top | 60 (26.3%) | 15 (23.1%) | 45 (27.6%) |
| | Top and bottom (versatile) | 58 (25.4%) | 19 (29.2%) | 39 (23.9%) |
| | More top than bottom | 49 (21.5%) | 12 (18.5%) | 37 (22.7%) |
| | Top only | 30 (13.2%) | 10 (15.4%) | 20 (12.3%) |
| | Not stated | 3 (1.3%) | 3 (4.6%) | 0 (0.0%) |
| **Number of anal sex partners in the past six months (N, %)** | | | | |
| | None | 14 (6.1%) | 0 (0.0%) | 14 (8.6%) |
| | 1 | 22 (9.6%) | 2 (3.1%) | 20 (12.3%) |
| | 2 to 5 | 67 (29.4%) | 12 (18.5%) | 55 (33.7%) |
| | 6 to 10 | 44 (19.3%) | 13 (20.0%) | 31 (19.0%) |
| | More than 10 | 76 (33.3%) | 35 (53.8%) | 41 (25.2%) |
| | Not stated | 5 (2.2%) | 3 (4.6%) | 2 (1.2%) |
| **Number of anal sex partners without using condom in the past six months (N, %)** | | | | |
| | None | 62 (27.2%) | 6 (9.2%) | 56 (34.4%) |
| | 1 | 54 (23.7%) | 7 (10.8%) | 47 (28.8%) |
| | 2 to 5 | 71 (31.1%) | 23 (35.4%) | 48 (29.4%) |
| | 6 to 10 | 21 (9.2%) | 11 (16.9%) | 10 (6.1%) |
| | More than 10 | 17 (7.5%) | 15 (23.1%) | 2 (1.2%) |
| | Not stated | 3 (1.3%) | 3 (4.6%) | 0 (0.0%) |

STI, sexually transmitted infection

*or similar

**for example A levels, high school diploma, German "Abitur".

demographic and sexual risk behaviour data for these respondents (analysis sample), as well as separately for those who were using PrEP (or had a history of its use) and those who were considering it. Differences between these latter two subgroups were minor and similar to the roughly analogous subgroups reported in the publication on the quantitative results of the survey [30].

See S1 Table for a comparison of the demographic data and sexual risk behaviour of participants in our analysis sample and respondents who indicated that they were neutral about or might be interesting in taking PrEP but did not provide a free-text answer to the question about their motivation (n = 104). The latter group was very similar to our analysis sample in terms of age, educational attainment and place of residence, but was different to it insofar as a substantially larger proportion of respondents were less financially secure, were unsure about their HIV status, had never used PrEP and reported engaging in sexual behaviour that put them at lower risk of HIV infection.

## Motivations of the participants

The number of motivations cited by participants ranged from one to five (mean: 1.38, median: 1.0). More specifically, of the 228 responses to the question about participants' motivation for

**Table 3. Participants reporting one or more motivations exclusively in a category or combination of categories in the analysis sample, by subgroup (using or considering PrEP).**

| Category of motivation(s) for using or considering PrEP | Participants in analysis sample | Participants using PrEP (or with history of its use) | Participants considering PrEP use |
|---|---|---|---|
| | N = 228 | N = 65 | N = 163 |
| Safety | 121 (53.1%) | 30 (46.2%) | 91 (55.8%) |
| Mental well-being | 23 (10.1%) | 7 (10.8%) | 16 (9.8%) |
| Condom attitudes | 21 (9.2%) | 4 (6.2%) | 17 (10.4%) |
| Expectations | 14 (6.1%) | 4 (6.2%) | 10 (6.1%) |
| Norms | 2 (0.9%) | 1 (1.5%) | 1 (0.6%) |
| **TOTAL RESPONDENTS** | **181** | **46** | **135** |
| Safety & mental well-being | 12 (5.3%) | 4 (6.2%) | 8 (4.9%) |
| Safety & condom attitudes | 10 (4.4%) | 4 (6.2%) | 6 (3.7%) |
| Safety & expectation | 6 (2.6%) | 5 (7.7%) | 1 (0.6%) |
| Expectations & mental well-being | 6 (2.6%) | 3 (4.6%) | 3 (1.8%) |
| Expectations & condom attitudes | 4 (1.8%) | 1 (1.5%) | 3 (1.8%) |
| Mental well-being & condom attitudes | 4 (1.8%) | 1 (1.5%) | 3 (1.8%) |
| **TOTAL RESPONDENTS** | **42** | **18** | **24** |
| Safety, mental well-being & condom attit. | 4 (1.8%) | 1 (1.5%) | 3 (1.8%) |
| Safety, expectations & mental well-being | 1 (0.0%) | 0 (0.0%) | 1 (0.6%) |
| **TOTAL RESPONDENTS** | **5** | **1** | **4** |

using or considering PrEP, we coded 152 as describing one motivation, 66 as describing two motivations, 9 as describing three motivations, and 1 as describing five motivations. All free-text responses that contained multiple motivations reported these clearly as part of a list using the conjunctions "and" or "but", a comma or other form of punctuation, or some combination of these.

Table 3 gives an overview of the different categories of motivations and the number of respondents who reported one or more motivations exclusively in each category (or combination of categories). Of the 228 respondents, 154 reported some form of safety and protection against HIV as either their only motivation (n = 121) or one of their motivations (n = 33) for using or considering PrEP. This was followed by 50 respondents who reported some form of mental well-being and quality of life as either their only motivation (n = 23) or one of their motivations (n = 27). A total of 43 respondents reported condom attitudes (i.e., a desire or intention to engage in condomless sex) as either their only motivation (n = 21) or one of their motivations (n = 22). Lastly, 31 respondents reported some form of expectations about sexuality as either their only (n = 14) or one of their motivations (n = 17). Only two respondents reported norms and social perspectives as their only motivation for using or considering PrEP. Among those respondents who reported two motivations in total, the only combinations of motivations that comprised at least approximately 5% of the analysis sample were in the categories safety and mental well-being (12 respondents, 5.3%) and safety and condom attitudes (10 respondents, 4.4%). The difference in the frequencies between the two subsamples was not statistically significant (Fisher's exact test, p = 0.234). A post hoc pairwise comparison of the frequencies also revealed that none of them differed in a statistically significant manner between the groups when the alpha level was set to 0.003 (i.e., 0.05/13).

Looking at the subcategories of motivations adds more granular information and complexity to this picture. In the safety category, responses citing general aspects of safety and

protection against HIV predominated (mentioned by 40.4% of respondents), but were followed by more specific motivations, such as PrEP serving as additional protection above and beyond condom use (16.2%), being at risk of HIV infection (7.5%), or seeking protection against HIV when not using condoms (6.1%). Nine respondents (0.04%) mentioned safety or protection in relation to protecting others as opposed to themselves. In the condom attitudes category, 18.0% percent of respondents explicitly mentioned a desire to engage in condomless sex as a motivation for using or considering PrEP. Expectations about sexuality, as a category, included explicit mentions of anxiety- or worry-free sex (8.3%) and of more pleasurable sex (6.1%) as motivations. Lastly, the category of mental well-being and quality of life included motivations related to reducing anxiety, fear or worries of being infected with HIV (8.6%) and a desire to increase quality of life or sexual and personal freedom (8.3%). Table 4 gives

**Table 4. Qualitative results: Survey respondents' motivations for using or considering PrEP according to category and subcategory, their frequency, and representative examples.**

| CATEGORY: SAFETY/PROTECTION AGAINST HIV | | |
|---|---|---|
| Subcategory | Frequency | Example motivations |
| General safety/Protection against HIV/Prevention of HIV | 92/228 (40.4%) | "To protect myself from HIV" |
| | | "Reduce the chance of HIV infection" |
| | | "Safety regarding HIV infection" |
| | | "HIV prophylaxis" |
| | | "Health protection" |
| Additional protection against HIV | 37/228 (16.2%) | "Additional safety when having sex" |
| | | "Backup" |
| | | "I want extra protection in case a condom fails or if I make a bad decision." |
| | | "To protect myself from HIV if the condom slips off or loses its protection for some other reason" |
| | | "Protection (double, with condom)" |
| Being at self-perceived risk of HIV | 17/228 (7.5%) | "HIV-infected partner who has just begun therapy" |
| | | "It has happened in the past, even though I didn't intend to, that I had sex without a condom" |
| | | "Many casual sex partners from time to time" |
| | | "Protection, I live in a long-term relationship with an HIV positive guy" |
| Protection against HIV when not using condoms | 14/228 (6.1%) | "To feel safer in case of unprotected sex" |
| | | "Easy protection when having sex without a condom" |
| | | "Sex without a condom and hardly any risk of infection" |
| | | "Protection against HIV without a condom" |
| Autonomy and self-empowerment | 9/228 (3.9%) | "To protect myself more actively and not rely on others, for example regarding a condom" |
| | | "More autonomy as (I am) more a bottom" |
| | | "I have more safety and don't have to depend on my partner keeping the condom on" |
| | | "Because I find it hard to trust people, so as a way to be more careful." |
| | | "More control" |
| Protecting partner(s) or relationship(s) | 9/228 (3.9%) | "To protect health (mine, of my partner & of my sex partner)" |
| | | "To protect my opposite" |
| | | "Protection for me and others" |
| | | "Greater security in an open relationship" |
| | | "The knowledge that I can't harm anybody with my actions" |
| | | "My partner has HIV and we're in an open relationship" |
| Protection against HIV during periods of anticipated increased risk (e.g. recreational drug use, holidays) | 5/228 (2.2%) | "To avoid accidents when I'm in environments that may affect my decisions and behaviour. At a party or around the time I go for a party. I am afraid that alcohol consumption or simply condom break may expose me to HIV." |
| | | "Addition protection during special occasions (e.g., holiday)" |
| | | "I have more security and don't have to rely on (. . .) myself, even under the influence of alcohol, insisting on a condom in every situation" |
| | | "Protection against stupidity when drinking alcohol" |
| PrEP as an affordable protection against HIV | 2/228 (0.0%) | "Protection, low costs" |
| | | "It seems like an affordable option and viable way to help protect against HIV infection." |
| CATEGORY: MENTAL WELL-BEING AND QUALITY OF LIFE | | |
| Subcategory | Frequency | Example motivations |

*(Continued)*

**Table 4.** (Continued)

| | | |
|---|---|---|
| Reducing anxiety, fear or worries of being infected with HIV | 17/197 (8.6%) | "Paranoia, fear of getting infected" |
| | | "Not constantly having to be afraid" |
| | | "Less anxiety before the next HIV test" |
| | | "My whole life there's only been sex with a condom and fear of HIV" |
| | | "To free myself from fear" |
| | | "Although I'm putting myself at risk of getting an STD, I find the benefits of PrEP overwhelmingly because I no longer have to fear that I'll get HIV" |
| Desire to increase quality of life or sexual / personal freedom | 19/228 (8.3%) | "Peace of mind" |
| | | "Convenience" |
| | | "Quality of life" |
| | | "Not always (. . ..) having to take PEP" |
| | | "Personal freedom" |
| | | "Sexual freedom" |
| | | "A feeling of security" |
| | | "Uncomplicated sex" |
| | | "Spontaneous sex (. . .) also with casual partners" |
| | | "I'd like to try a few sex partners who I otherwise couldn't (try out) without being on PrEP" |
| | | "So I can behave more like heterosexuals and not worry every time I choose not to use a condom" |
| Reducing periods of anticipated regret | 8/228 (3.5%) | "The psycho-stress after unprotected sex" |
| | | "No guilty conscience about unsafe sex" |
| | | "To not feel regret after unsafe sex" |
| | | "Less chance for my imagination to run away from me" ("weniger Kopfkino") |
| | | "To have a better conscience after having unsafe sex" |
| Desire for a healthy life | 7/228 (3.1%) | "To protect (my) health" |
| | | "To not become ill" |
| | | "Longer life" |
| | | "Health" |
| | | "Stay healthy" |

| CATEGORY: CONDOM ATTITUDES | | |
|---|---|---|
| **Subcategory** | **Frequency** | **Example motivations** |
| Desire or intent to engage in condomless sex | 41/228 (18.0%) | "I don't like condoms" |
| | | "Anal sex without a condom" |
| | | "Unsafe sex" |
| | | "Unprotected sex" |
| | | "To have sex without condom with known sexual partners that test for other STDs regularly" |
| | | "Unprotected sex with partner" |
| | | "To have riskier sex" |
| Difficulties with condom use | 2/228 (0.9%) | "Protection from HIV because I can't deal with condoms" |
| | | "Problem with condom when being a top" |

| CATEGORY: EXPECTATIONS ABOUT SEXUALITY | | |
|---|---|---|
| **Subcategory** | **Frequency** | **Example motivations** |
| Expectations of worry-free sex | 19/228 (8.3%) | "It's a way to feel safer when having sex" |
| | | "More relaxed approach to sexuality" |
| | | "Unencumbered Sex" |
| | | "Sex without fear" |
| | | "To have riskier sex without fear" |
| | | "Carefree sex without worrying about HIV infection" |
| Expectations of more pleasureful sex | 14/228 (6.1%) | "More pleasure" |
| | | "Sex is more intense" |
| | | "To enjoy sex" |
| | | "To (. . .) enjoy sex more" |
| | | "Intense feeling during sex" |
| | | "More sensuality, more pleasure" |

| CATEGORY: NORMS / SOCIAL PERSPECTIVES | | |
|---|---|---|
| **Subcategory** | **Frequency** | **Example motivations** |
| Perceiving condomless sex/PrEP intake as a social norm | 1/228 (0.4%) | "More and more guys are doing bareback sex only" |
| Desire to eradicate HIV | 1/228 (0.4%) | "Eradicating HIV" |

examples of the motivations ordered along these categories; responses in German were translated into English for presentation in the table by one researcher (MG) and checked independently by the remaining authors, with disagreements resolved by consensus.

## Discussion

Our study is the first multicentre, paper-based survey to ask MSM in Berlin explicitly to name their motivations for using or considering HIV PrEP. By a large margin, most respondents who answered the question cited some form of safety and protection against HIV as their only motivation or one of their motivations in this regard. Most frequently in this category, respondents mentioned being motivated by the idea of having additional protection in case of condom failure (e.g., "To protect myself from HIV if the condom slips off or loses its protection for some other reason", "Additional safety when having sex"). Even though some 18% of respondents in our analysis sample explicitly mentioned a desire or intent to engage in condomless sex as a motivation for using or considering PrEP, these findings about safety and protection should reassure payers and practitioners who might be focusing disproportionately on the subject of sexual risk compensation, i.e., an increase in sexual behaviours that could put those who engage in them at a higher risk of infection with HIV and other STIs [75–80]. Indeed, the second most frequently cited category of motivations among our respondents after that of safety and protection was that of mental well-being and quality of life. Given the alarming frequency of suicide, depression and self-harm among sexual minorities, for which there is an abundance of evidence worldwide (e.g., [81–83]), our results suggest that it might be wise to attach greater weight to these beneficial effects of PrEP when deciding on its appropriateness for a given patient–and perhaps somewhat less weight than hitherto to the understandable but still largely theoretical concern of increased sexual risk compensation and STI incidence, for which the evidence is mixed [84–86].

This latter finding resonates with the results of qualitative studies that have engaged with this area of enquiry, challenging the predominant biomedical approach to PrEP implementation. With regard to motivations such as those to reduce HIV anxiety, improve self-efficacy, be able to make one's own informed choices, and feel less fear and shame, for example, our respondents wrote short but powerful responses about "not constantly having to be afraid," living a "whole life" in which "there's only been sex with a condom and fear of HIV", wanting to "try a few sex partners who I otherwise couldn't without being on PrEP", and no longer having to "feel regret" or have "psycho-stress" after having "unprotected" or "unsafe" sex. Examining such phenomena, and affective experiences more generally, is crucial because these have been shown to have a profound impact on the sexual and psychological well-being of MSM (e.g., [35, 37, 38, 57, 59, 60]). Moreover, these motivations are also tied to ongoing decision-making about PrEP that is based on sexual risk and relationship trajectories that have been shown to be varied and dynamic [37].

This idea of moving beyond a circumscribed biomedical approach to PrEP implementation is supported by a further, and somewhat unexpected, result from our study: Even though our survey item asked only for respondents' main motivation for using or considering PrEP, one third (76/228) of our respondents nevertheless reported more than one motivation. Although these multiple motivations often fell into the same category (most frequently into that of safety and protection), it is important to bear in mind that even in a category as seemingly one-dimensional as "safety and protection", a broad range of subcategories could be identified, including protection during periods of anticipated increased risk, self-perceptions of being at high risk, and additional protection against HIV while using a condom. Moreover, some 20% (47/228) of respondents reported motivations in more than one category. Altogether, this

suggests that approaches to PrEP counselling that seeks, where appropriate, to increase the uptake of PrEP by focusing exclusively on its effectiveness in preventing HIV is unlikely to be as successful as a holistic approach that focuses simultaneously on multiple motivating factors, particularly those related to mental health and quality of life. While this may be something that any good doctor, sexual health counsellor or social worker would do, the results of our study might nevertheless be useful as a reminder of the breadth of motivations that can influence the decision to use PrEP and encourage practitioners to seek to identify multiple (and, indeed, even apparently contradictory) factors in individual patients. This may also aid decisions to inform patients about other preventive strategies, and to provide support with broader sexual and psychological health issues, as part of a more holistic approach.

The idea of taking a more holistic approach to PrEP implementation has been the focus of several recent publications. Lacombe-Duncan et al. (2021) drew on qualitative data from in-depth, semi-structured interviews with 29 MSM recruited purposively in 2015/16 in a community setting in Toronto, Canada, to identify and explore gaps in PrEP implementation that might be amenable to social work intervention [61]. Among other results, they found that MSMs' navigation of sexual health and risk practices was made complex by a range of sometimes contradictory factors encompassing the individual, interpersonal, organisational and structural levels. This aligns very much with the breadth of motivations reported in our survey, as well as the reporting of multiple motivations by individual respondents. Lacombe-Duncan et al. (2021) make clear that addressing such factors and, more specifically, the social determinants of health and psychosocial issues that affect PrEP engagement solely in terms of HIV infection risk and a binary decision about PrEP uptake would be wholly inadequate to support individuals with informed decision making [61]. A recently published systematic review by Pinto et al. (2018) corroborates these and, in part, our findings, and identifies the need for a range of multilevel interventions targeting multiple socioecological domains, such as interventions to help people navigate health care systems and improve referrals to mental health and supportive services, as well as provider-level interventions such as increased training and education about PrEP [43]. In their scoping review of PrEP service delivery and programming, Hillis et al. (2020) also identify a range of potentially useful multidisciplinary and innovative PrEP care pathways; however, they make the important additional point that PrEP provision in a health system actually creates new opportunities for MSM to access health services to which they might not otherwise have availed themselves, such as sexual health care, testing, treatment, counselling and STI testing and psychological support [87].

A further aim of our study was to determine whether motivations differed between individuals at two different locations on a conceptual continuum of PrEP care: those who were using it or had used it at some point in the past and those who were contemplating or considering its use. Contrary to our expectations, we were unable to identify statistically significant differences in motivations between the groups that might inform targeted approaches to increase uptake or improve adherence. From a qualitative research perspective, however, it is of value to point out that some of the differences appeared substantial and would be worthy of further investigation, ideally applying methods and a study design that would allow for quantitative hypothesis testing as well as in-depth qualitative questioning of findings. In particular, it appears that participants considering PrEP may have been more likely to cite safety as their only motivation whereas motivations in combination with the category of safety played a greater role among those using PrEP or with a history of its use. While this may be due to social desirability bias in survey research, it is conceivable that once individuals have initiated PrEP, other categories of benefits–for example related to expectations around sexuality or mental well-being–may become more readily apparent to them. Interestingly, the comprehensive review of the literature undertaken by Young et al. (2013) before the wide rollout of PrEP found that willingness

to take PrEP (the closest construct to motivations that they examined) was associated with younger age, unprotected anal intercourse with casual partners and increased risk perceptions; on the other side, anxieties around medical side effects of PrEP were commonly reported in the included studies as barriers. Psychosocial correlates of willingness, such as reduced anxiety about HIV infection or an increase in self-efficacy, do not appear to have been reported as frequently in this pre-rollout phase, however. While this may simply be an artefact of the biomedical approach to these questions taken by many of the studies included in their review, it might also support the hypothesis that these positive aspects of PrEP only become more readily apparent after PrEP use has been initiated.

Some of our findings were surprising. Attitudes towards using PrEP have been shown in the literature to be linked to considerations of protecting primary and/or outside partners [73], and concerns for sexual partners' risk of acquiring HIV and the public health have also been described as a facilitator of engaging in safer sex practices [74]. In our study, however, among the high number of respondents who reported some form of safety or protection against HIV as their only motivation or as one of their motivations (n = 154), only a few (n = 9) cited protecting their partners or relationships as a relevant motivational factor. One reason for this may be that a requirement for initiating PrEP, unlike using a condom, is that an individual gets tested for HIV. Knowing with greater certainty that they were not infected with HIV may have led them to focus more on the protective effects of PrEP for themselves rather than the thought of preventing onward transmission. Regardless, given the evidence that individuals are more averse to risk when their decisions affect others [74, 88, 89], focusing on risks to others to convince people to engage in beneficial public health behaviours remains an important part of public health messaging and initiatives. Moreover, the literature [37, 80, 90, 91] and anecdotal reports from the centres that took part in our study suggest that, increasingly, PrEP use is becoming a social norm and many individuals feel pressured to use PrEP in a social environment that insists on sex without condoms. Surprisingly, however, only one participant in our study mentioned this as a motivation for using or considering PrEP ("More and more guys are doing bareback sex only"). The reasons for this result are unclear, but may be related to an already long established culture of unprotected sex in subgroups of the gay population in large cities like Berlin, or to the fact that, at the time of our survey, PrEP had only just become available through formal channels in Germany. Regardless, the topic warrants further research, particularly of a qualitative nature.

An important strength of our survey is that we asked participants explicitly about their motivation for considering or using PrEP rather than interpreting various factors correlated with PrEP contemplation or PrEP use as motivations. Another strength is its facility-based, multicentre design and paper format, which may have avoided or ameliorated some of the disadvantages of online surveys, such as self-selection bias and multiple responses from the same individuals. Nevertheless, our study also has a number of substantial limitations beyond its cross-sectional, observational design and the obvious caveats that this entails. First, the results of survey-based studies are subject to social desirability bias [92–96]. It is probable that some of our respondents indicated that their motivation for taking PrEP was safety because they wanted to project a favourable image of themselves or, despite the anonymous nature of the survey, they thought doing so would ensure continued good care from their physician or make it more likely they would obtain a prescription for PrEP. However, we found no statistically significant differences between the motivations reported by respondents who had a history of PrEP use and those who were considering it. Presumably individuals who had already taken PrEP before will have had fewer concerns about gaining access to the medication. Moreover, if reporting socially desirable motivations had been a strong force behind respondents' answers, one might have expected, from both subgroups, a larger number of responses focusing on the

protection of others or the public health more generally. Second, although our coding framework was developed and implemented in a systematic manner by three independent researchers, any set of codes and categories will always, to a certain extent, be arbitrary. We attempted to address this point by testing the assumption of independence of observations in larger and broader categories (as opposed to our subcategories), but it is possible that a different system of coding would yield different results. Third, due to our study being based on a paper survey with limited space for a free-text response, we were unable to pursue an important avenue of qualitative enquiry, namely in-depth follow-up questions leading from the general to the specific. Without this, it is impossible to know what the more superficial but also the deeper motivations for using or considering PrEP might be. In-depth interviews exploring our findings in future projects might bring more clarity. Fourth, respondents to our survey who indicated that they were neutral about or might be interested in taking PrEP but did not respond to our open question may have had different motivations than those who did provide a response. Our informal comparison of these two groups (see S1 Table) suggests that these individuals may have been less interested in PrEP because their self-perceived risk of HIV infection was lower due to their sexual risk behaviour (e.g., fewer anal sex partners and more condom use). Moreover, the findings of our study and the larger quantitative survey of which it is a part [30] are specific to MSM in Berlin and therefore limited in their generalisability. Nevertheless, they can provide a helpful comparison to the situation in cities with roughly similar populations of MSM such as London, San Francisco, Paris or New York, where the implementation of PrEP is already well underway. Fifth, because of space limitations in our paper-based survey we did not ask for explicit motivations for not using PrEP. Doing so would have added an interesting dimension to the study and allowed us to explore additional steps along the PrEP care continuum and approaches to promoting alternative prevention strategies. Sixth, although we did not exclude transgender MSM from taking part in our survey, we did not explicitly instruct participating centres to include or exclude this group. Anyone who identified as male (cis or trans) and reported having sex with other MSM could take part. Other sampling strategies could have been used to obtain meaningful data on transgender MSM's motivations for using or considering PrEP but would have gone beyond the scope of our study.

## Conclusion

As part of a broader multicentre survey of MSM in Berlin, Germany, we additionally asked participants in an open-ended question about their main motivation for using or considering the use of PrEP. While the focus of their responses lay on safety, the range of motivations was broad and could be grouped into five main categories, listed in descending order of frequency: safety and protection again HIV, mental well-being and quality of life, a desire or an intention to engage in sex without a condom, expectations about sexuality, and norms and social perspectives. There were no statistically significant differences in motivations between participants who were using PrEP already versus those who were considering its use. Many respondents reported several motivations, and responses overall were heterogeneous. This suggests that health professionals seeking, where appropriate, to increase PrEP uptake by focusing exclusively on its effectiveness in preventing HIV is unlikely to be as successful as a holistic approach that focuses on multiple motivating factors, particularly those related to mental health and quality of life. These results may inform health providers' approach to PrEP education and prescribing, as well as the design of information campaigns and other interventions to increase PrEP uptake alongside other strategies of HIV prevention and support for the broader sexual and psychological health of MSM.

## Supporting information

**S1 Table. Demographic data and sexual risk behaviour of participants in analysis sample and participants who were neutral about or potentially interested in taking PrEP but who did not answer the question about their main motivation.**
(DOCX)

**S1 Dataset. Minimal underlying data set and codebook.** Age of respondents has been removed to ensure patient anonymity.
(XLSX)

## Acknowledgments

The authors would like to acknowledge and thank the following centres and organisations Gemeinschaftspraxis Dietmar Schranz und Klaus Fischer and staff; Praxiszentrum Kaiserdamm and staff; Praxis Wünsche and staff; Novopraxis Berlin GbR and staff; Praxis Jessen[2] + Kollegen and staff; Ärztezentrum Nollendorfplatz and staff; Mann-O-Meter e.V. and staff; Pluspunkt Berlin (Schwulenberatung Berlin gGmbH) and staff; Fixpunkt e.V. and staff; Berliner AIDS-Hilfe e.V. and staff for their participation in distributing questionnaires and collecting data. The study would not have been possible without the participants who consented to be part of the study. We acknowledge open access publishing support from the German Research Foundation (DFG) and the Open Access Publication Fund of Charité – Universitätsmedizin Berlin.

## Author Contributions

**Conceptualization:** Matthew Gaskins, Frank Kutscha, Alexander Nast, Ricardo Niklas Werner.

**Formal analysis:** Matthew Gaskins, Mary Katherine Sammons, Ricardo Niklas Werner.

**Investigation:** Matthew Gaskins, Ricardo Niklas Werner.

**Methodology:** Matthew Gaskins, Mary Katherine Sammons, Frank Kutscha, Ricardo Niklas Werner.

**Project administration:** Matthew Gaskins, Ricardo Niklas Werner.

**Supervision:** Alexander Nast, Ricardo Niklas Werner.

**Writing – original draft:** Matthew Gaskins.

**Writing – review & editing:** Matthew Gaskins, Mary Katherine Sammons, Frank Kutscha, Alexander Nast, Ricardo Niklas Werner.

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
