## [Decision Letter · Decision Letter 0]

22 Jul 2021

PONE-D-21-17716

Factors that motivate men who have sex with men in Berlin, Germany, to use or consider using HIV pre-exposure prophylaxis – A mixed-methods analysis of data from a multicentre survey

PLOS ONE

Dear Dr. Gaskins,

Thank you for submitting your manuscript to PLOS ONE. After careful consideration, we feel that it has merit but does not fully meet PLOS ONE’s publication criteria as it currently stands. Therefore, we invite you to submit a revised version of the manuscript that addresses the points raised during the review process.

Both reviewers found the paper interesting and valuable, but they raised a number of important issues that need to be addressed. These include clarifying the study background and reframing the rationale; referencing and addressing a number of relevant studies that are omitted (in Intro and Discussion); more fully describing the methods and data analysi; and providing context on PrEP availability in Germany and the broader EU context for international readers.

We look forward to receiving your revised manuscript.

Kind regards,

Peter A Newman, Ph.D

Academic Editor

PLOS ONE

Journal Requirements:

2. Please include additional information regarding the survey or questionnaire used in the study and ensure that you have provided sufficient details that others could replicate the analyses. For instance, if you developed the survey or questionnaire as part of this study and it is not under a copyright more restrictive than CC-BY, please include a copy, in both the original language and English, as Supporting Information. If the questionnaire is published, please provide a citation to the (1) questionnaire and/or (2) original publication associated with the questionnaire.

Reviewers' comments:

Reviewer's Responses to Questions

**Comments to the Author**

1. Is the manuscript technically sound, and do the data support the conclusions?

Reviewer #1: Yes

Reviewer #2: Yes

2. Has the statistical analysis been performed appropriately and rigorously? 

Reviewer #1: Yes

Reviewer #2: Yes

3. Have the authors made all data underlying the findings in their manuscript fully available?

Reviewer #1: Yes

Reviewer #2: Yes

4. Is the manuscript presented in an intelligible fashion and written in standard English?

Reviewer #1: Yes

Reviewer #2: Yes

5. Review Comments to the Author

Reviewer #1: Thank you for the opportunity to review “Factors that motivate men who have sex with men in Berlin, Germany, to use or consider using HIV pre-exposure prophylaxis – A mixed-methods analysis of data from a multi-centre survey”. The authors conducted a thorough analysis of both quantitative and qualitative data from a cross-sectional survey conducted with a relatively large sample of MSM with complete data on the qualitative component (n=228). The manuscript contributes to the broader literature about the importance of understanding social and structural factors in PrEP uptake. The authors’ conclusion that health professionals seeking to increase PrEP uptake need take a holistic approach that emphasizes multiple motivations is well-founded in their data. My enthusiasm for this manuscript is tempered by three major concerns. Specifically, the manuscript is in need of: 1) a more thorough literature review that draws on existing research about motivations for PrEP use; 2) clarification of the mixed-methods methodology; and 3) re-structuring of the methods/results. In addition to providing more details regarding these concerns, I offer some additional feedback by manuscript section below for the authors’ consideration.

Abstract

I’m not completely clear based on the results presented in the abstract how recognition of multiple and varied motivations may reassure practitioners concerned about condom use in relation to PrEP, particularly when the authors note that almost 1/5 intended not to use condoms. I would consider removing that line, as it requires too much context to be flushed out in an abstract in a nuanced way, but is explored in more depth in the discussion section of the manuscript.

I’m unclear exactly what the authors mean by ‘categories’ in the sentence “…responses overall were heterogeneous, even within categories”; additional clarification would be most helpful.

I appreciate the last line of the conclusion as a clear message for practitioners that is well-founded in the data.

Introduction

The authors strong opening paragraphs which detail the background of PrEP approval, the importance of PrEP to reducing HIV disparities particularly among MSM, and context of PrEP uptake in Berlin, Germany. In paragraph 2, it would be helpful for the authors to hypothesize or provide data pertaining to why those who were likely to use PrEP were not able to access it. I would also be interested to know a bit more about why policy makers in many countries are debating integration of PrEP into publicly-funded systems, but recognize that this may be beyond the scope of the paper.

In between the lines starting “As of late 2019…” and “This is unfortunate…” it would be helpful to add one line and associated references that refers to the disproportionate prevalence of HIV among MSM in the US context (see: https://www.cdc.gov/hiv/group/msm/index.html).

In terms of flow, I would consider switching the order of paragraphs 3 on the PrEP continuum of care and paragraph 4 on barriers to PrEP uptake.

I appreciate the authors’ attentiveness to social and structural barriers to PrEP uptake and considerations of a PrEP continuum of care. To that end, there is some literature that should be integrated into the PrEP continuum of care paragraph as well as used to contextualize the discussion. Please see:

https://www.ncbi.nlm.nih.gov/pmc/articles/PMC6253066/

https://www.ncbi.nlm.nih.gov/pmc/articles/PMC5315535/

https://journals.lww.com/jaids/Abstract/2021/04150/The_PrEP_Cascade_in_a_National_Cohort_of.4.aspx?context=LatestArticles

While the latter two articles follow more biomedically-oriented conceptualizations of a PrEP cascade, the first article (Newman et al., 2018) details social and structural considerations of PrEP uptake across a PrEP continuum of care, some of which overlaps with the findings of this manuscript. Moreover, given the topic of the paper is on motivations of PrEP use, I believe that the sentence “However, our knowledge about explicit motivations that influence decisions to initiate PrEP is less well developed and stems mostly from small qualitative studies involving focus groups or interviews [38-42]” warrants it’s own paragraph and more details. There is also additional research on motivations for PrEP use, some of which corroborates the findings of the researchers, that could be integrated into the introduction and the discussion, for example:

https://sti.bmj.com/content/95/4/262.abstract

https://academic.oup.com/abm/article/49/2/177/4561523?login=true

Then, it may be helpful to add one additional statement explicitly summarizing gaps in the evidence base.

Minor notes: line 78 at first mention of United States include US in brackets, as you use US throughout after that; also, MSM should be spelled out at first mention.

Materials and Methods

I appreciated the thorough overview of recruitment, and in particular the in-depth description of the non-municipal counselling and testing centers and HIV specialist services.

Was there a rationale for randomly selecting 30 out of 228 responses to create the coding framework? For example, did the authors reach some level of saturation (no new category generation) by open-text response 30? Upon coding the remaining free-text responses, were there any motivations that did not fit within the originally-developed framework? If so, how did the authors resolve this?

In terms of structure, I believe it would be clearer if the sub-heading ‘Literature search’ was removed and the literature search paragraph integrated into the analysis section before the line “These two classifications systems…”. Moreover, the operationalization of the PrEP care continuum is related to the quantitative analysis and could be absolved into that section before the line beginning with “For any post-hoc pairwise comparisons…”

Table 1: I appreciate the authors’ integration of references. Some of the suggested references above may also be able to be integrated into the table so that there are references for most of the sub-categories (E.g., Newman et al. (2018) could be used to reference ‘expectations of more pleasureful sex or increased intimacy and closeness (when not using a condom)’, ‘ expectations of worry-free sex’, ‘desire to increase quality of life or sexual/personal freedom’, ‘reducing anxiety, fear, or worries of being infected with HIV’, and ‘autonomy and self-empowerment in the protection against HIV’, among other sub-categories.

One aspect I wondered about in the Abstract, and now here, is the potential overlap between ‘worry-free sex’ and ‘reducing anxiety, fear, or worries of being infected with HIV’ – can the authors expand more on how they are conceptualizing the difference between these sub-categories throughout the manuscript.

For clarity, I suggest using the sub-headings ‘Qualitative data analysis’ (for section that is now called Data analysis) and then ‘Quantitative data analysis’ (for the section that is now called Sample size and statistical methods).

The way that the study is currently depicted is as a multi-method as opposed to a mixed-methods study. Mixed methods is an in-depth methodology that goes beyond the rigorous collection and analysis of both quantitative and qualitative data to include the integration of the two forms of data using a pre-specified mixed methods typology (See: Creswell & Plano-Clark, 2017), thus, more details about the mixed-methods methodology should be included. As the key to mixed-methods is integration of quantitative and qualitative data, if mixed-methods is maintained as the methodology, another section should be added titled ‘Mixed methods analysis’ that details how quantitative and qualitative data were compared/contrasted (See: Fetters, Curry, & Creswell, 2013). Alternately, the authors may opt to refer to their study as a multi-method study

Minor: Please spell in full at first use of LGBTI+.

Results

The first paragraph of the results could be integrated within the methods section ‘Sampling methods, participants, and settings’, up until the last line; the last line could be included in the section ‘Demographic data, sexual behavior, and HIV risk of analysis sample’.

Table 2 – rows with 0 responses (e.g., primary education, small town/rural area in Germany) could be omitted.

The lines “There was a high degree of agreement between the coding results…” should be integrated within the qualitative data analysis sub-section. The subsection coding of responses could be more appropriately titled Motivations of Participants (or something similar). I don’t think the line “there were no disagreements between the reviewers in this regard” is needed – if maintained, it should also be moved to the analysis.

I may have missed it, but I’m not clear how the authors classified a motivation as a main motivation – does this just mean that the participant wrote it down as a motivation, or did the authors denote some order as to what was main when participants indicated more than one motivation? Please clarify.

Discussion

I am wondering if the authors have more context about provider concern about condom use as leading to hesitancy about PrEP – is this an issue in the EU context that has prevented widespread adoption of PrEP into public insurance plans? Additional references would be helpful.

The authors may also consider a broader discussion of literature pertaining to PrEP counselling from a holistic lens, as they are describing, see:

https://academic.oup.com/hsw/article-abstract/46/1/22/6151208

https://link.springer.com/article/10.1007/s10461-020-02855-9

https://www.tandfonline.com/doi/abs/10.1080/09540121.2020.1744507

Excellent overview of the limitations of the study.

Reviewer #2: This paper reviews the motivations for PrEP use among a sample of MSM living in Berlin. It is methodologically sound and well-written with some thought-provoking conclusions, particularly with respect to the mental health circumstances of MSM and how these may be alleviated, for some, by the use PrEP. There are, however, a few issues that I think need to be addressed before it is ready for publication.

1. Recognising and reflecting previous research. The emergence of PrEP has seen a significant number of papers published that examine: the acceptability of PrEP; motivations for PrEP use; factors that may influence PrEP uptake and continued use; how PrEP is integrated into sexual lives; how PrEP reforms how MSM think about HIV risk, etc. As the key rationale for the study, the authors states that most prior research has been focus group or interview based and, therefore, larger scale data is required. I’m not sure this accurately reflects the nature of the literature on this topic (where there are a number of quantitative or survey-based studies) nor does being ‘bigger’ necessarily constitute a clear rationale for this study. I would encourage the authors to acknowledge the large amount of quantitative literature on this topic (look, for example, at work by Bavinton, Prestage and Holt, among others) and consider recasting the principle rationale for the study to identify what is unique.

2. Acceptable in theory and in practice. Partially linked to the point above, prior to the widespread rollout of PrEP there were a large number of studies looking at the theoretical acceptability of PrEP (see review by Ingrid Young). What I found myself wondering was how the findings of your study – when people CAN actually access and use PrEP differ from/relate to the motivations MSM provided when talking about PrEP in the abstract. Can you reflect some of this literature in your discussion?

3. Coding frame. Can you provide a clear rationale as to why you created a coding frame from a review of the literature and applied it to your data, rather than working with the data in an inductive manner? I also think it would be much easier to follow if you describe the literature review process, and its purpose, prior to presenting the coding frame.

4. As a minor point, can you include a short summary as to the status of PrEP availability in Germany a the time of data collection? Where could it be obtained? How much did it cost? What prescribing criteria has to be met? Such information would aid interpretation from an international audience.

6. PLOS authors have the option to publish the peer review history of their article (what does this mean?). If published, this will include your full peer review and any attached files.

Reviewer #1: No

Reviewer #2: No

---

## [Author Response · Author response to Decision Letter 0]

8 Sep 2021

Dear Professor Newman,

Dear members of the Editorial Board,

Dear Reviewers,

We would like to thank the academic editor and each of the reviewers for their careful review of our manuscript and their thorough and very helpful suggestions for improving it. We feel that the manuscript has benefitted greatly as a result. Please find our replies to each of their points in the rebuttal letter we have uploaded, including details on how we have implemented their suggestions. Please note that all line numbers in the rebuttal letter refer to the file ‘Revised Manuscript with Track Changes’. (If the reviewers prefer that the line numbers refer to the clean file without track changes, we would be happy to send a revised point-by-point reply letter with these instead.)

Yours sincerely,

Matthew Gaskins

---

## [Decision Letter · Decision Letter 1]

4 Nov 2021

Factors that motivate men who have sex with men in Berlin, Germany, to use or consider using HIV pre-exposure prophylaxis – A multi-methods analysis of data from a multicentre survey

PONE-D-21-17716R1

Dear Dr. Gaskins,

We’re pleased to inform you that your manuscript has been judged scientifically suitable for publication and will be formally accepted for publication once it meets all outstanding technical requirements.

Kind regards,

Peter A Newman, Ph.D

Academic Editor

PLOS ONE

Additional Editor Comments (optional):

Reviewers' comments:

Reviewer's Responses to Questions

**Comments to the Author**

1. If the authors have adequately addressed your comments raised in a previous round of review and you feel that this manuscript is now acceptable for publication, you may indicate that here to bypass the “Comments to the Author” section, enter your conflict of interest statement in the “Confidential to Editor” section, and submit your "Accept" recommendation.

Reviewer #1: All comments have been addressed

Reviewer #2: All comments have been addressed

2. Is the manuscript technically sound, and do the data support the conclusions?

Reviewer #1: Yes

Reviewer #2: Yes

3. Has the statistical analysis been performed appropriately and rigorously? 

Reviewer #1: N/A

Reviewer #2: N/A

4. Have the authors made all data underlying the findings in their manuscript fully available?

Reviewer #1: Yes

Reviewer #2: Yes

5. Is the manuscript presented in an intelligible fashion and written in standard English?

Reviewer #1: Yes

Reviewer #2: Yes

6. Review Comments to the Author

Reviewer #1: Thank you for the opportunity to re-review the manuscript now titled “Factors that motivate men who have sex with men in Berlin, Germany, to use or consider using HIV pre-exposure prophylaxis – A multi-methods analysis of data from a multi-centre study”. The authors have addressed my comments to a high degree of quality. I agree with the authors' assessment that the expanded literature review leads into a clearer rationale for the study, both from a methodological and study site (geographic) perspective. The methodology is now clearer as described as a multi-methods study. The added description of qualitative analyses and restructuring of the methods (particularly moving the section previously entitled literature review) has also strengthened the clarity of the manuscript. I just have one minor comment: Where the authors quote Young & McDaid in the introduction a page number is needed.

Reviewer #2: I congratulate the authors on a really comprehensive revision of their manuscript. The integration of existing research, coupled with a reflection on how PrEP is perceived in practice (rather than just in theory) has helped to ensure a much more convincing rationale for the paper. The methodology is also much more clear now that the inductive-deductive process of data coding has been explained in more detail. It makes a useful addition to the literature on this topic.

7. PLOS authors have the option to publish the peer review history of their article (what does this mean?). If published, this will include your full peer review and any attached files.

Reviewer #1: No

Reviewer #2: **Yes: **Adam Bourne

---

## [Editor Report · Acceptance letter]

9 Nov 2021

PONE-D-21-17716R1 

Factors that motivate men who have sex with men in Berlin, Germany, to use or consider using HIV pre-exposure prophylaxis – A multi-methods analysis of data from a multicentre survey 

Dear Dr. Gaskins:

I'm pleased to inform you that your manuscript has been deemed suitable for publication in PLOS ONE. Congratulations! Your manuscript is now with our production department. 

Kind regards, 

on behalf of

Dr. Peter A Newman 

Academic Editor

PLOS ONE